# Genome-Wide Characterization and Analysis of R2R3-MYB Genes Related to Fruit Ripening and Stress Response in Banana (*Musa acuminata* L. AAA Group, cv. ‘Cavendish’)

**DOI:** 10.3390/plants12010152

**Published:** 2022-12-28

**Authors:** Zhuo Wang, Xiaoming Yao, Caihong Jia, Yunke Zheng, Qiumei Lin, Jingyi Wang, Juhua Liu, Zhao Zhu, Long Peng, Biyu Xu, Xinli Cong, Zhiqiang Jin

**Affiliations:** 1Key Laboratory of Tropical Crop Biotechnology of Ministry of Agriculture and Rural Affairs of China, Institute of Tropical Bioscience and Biotechnology, Chinese Academy of Tropical Agricultural Sciences, Haikou 571101, China; 2Hainan Academy of Tropical Agricultural Resource, Chinese Academy of Tropical Agricultural Sciences, Haikou 571101, China; 3Sanya Research Institute of Chinese Academy of Tropical Agricultural Sciences, Sanya 572024, China; 4BGI-Sanya, BGI-Shenzhen, Sanya 572025, China; 5College of Tropical Crops, Yunnan Agricultural University, Pu’er 665000, China; 6School of Life Sciences, Hainan University, Haikou 570228, China

**Keywords:** banana, R2R3-MYB, fruit ripening, stresses, WGCNA, expression analysis

## Abstract

MYB is an important type of transcription factor in eukaryotes. It is widely involved in a variety of biological processes and plays a role in plant morphogenesis, growth and development, primary and secondary metabolite synthesis, and other life processes. In this study, bioinformatics methods were used to identify the R2R3-MYB transcription factor family members in the whole *Musa acuminata* (DH-Pahang) genome, one of the wild ancestors of banana. A total of 280 *MaMYBs* were obtained, and phylogenetic analysis indicated that these MaMYBs could be classified into 33 clades with MYBs from *Arabidopsis thaliana*. The amino acid sequences of the R2 and R3 Myb-DNA binding in all MaMYB protein sequences were quite conserved, especially Arg-12, Arg-13, Leu-23, and Leu-79. Distribution mapping results showed that 277 *MaMYBs* were localized on the 11 chromosomes in the *Musa acuminata* genome. *The MaMYBs* were distributed unevenly across the 11 chromosomes. More than 40.0% of the *MaMYBs* were located in collinear fragments, and segmental duplications likely played a key role in the expansion of the *MaMYBs*. Moreover, the expression profiles of *MaMYBs* in different fruit development and ripening stages and under various abiotic and biotic stresses were investigated using available RNA-sequencing data to obtain fruit development, ripening-specific, and stress-responsive candidate genes. Weighted gene co-expression network analysis (WGCNA) was used to analyze transcriptome data of banana from the above 11 samples. We found *MaMYBs* participating in important metabolic biosynthesis pathways in banana. Collectively, our results represent a comprehensive genome-wide study of the *MaMYB* gene family, which should be helpful in further detailed studies on *MaMYBs* functions related to fruit development, postharvest ripening, and the seedling response to stress in an important banana cultivar.

## 1. Introduction

Transcription factors can specifically bind to cis-regulatory elements in the upstream promoter sequence of eukaryotic genes, regulate the gene expression through transcriptional activation or inhibition, and act as a switch for the expression of downstream genes. Transcription factors usually consist of four parts: the DNA binding domain, transcriptional activation domain, nuclear localization signal, and protein–protein binding domain [1]. MYB transcription factors comprise one of the largest families in eukaryotes [2]. Because of their large number and various functions, MYB transcription factors have always been a hot topic in functional studies in plants. The MYB transcription factor has a highly conserved MYB domain at the N-terminus. This domain generally consists of 51–53 amino acids and forms a helix-helix-turn-helix structure that interacts with target DNA [2,3]. Based on the number of MYB domain repeats, MYB proteins can be classified into four types: 1R-MYB, R2R3-MYB, R1R2R3-MYB, and 4R-MYB [2]. R2R3-MYB transcription factors, the largest subfamily of the MYB transcription factor family, are specific to plants and yeast [2,3]. The first identified MYB gene encoding the myb domain is COLORED1 (C1), which is necessary for anthocyanin synthesis in corn protein particles [4]. Based on their well-conserved DNA-binding domains, genome-wide identification of R2R3-MYB members has been conducted in various plants, such as Arabidopsis [3], *Oryza sativa* [5], *Vitis vinifera* [6], *Populus trichocarpa* [7,8], *Zea mays* [9], cotton [10], soybean [11], chinese cabbage [12], cucumber [13], tomato [14], *Medicago truncatula* [15], and wheat (*Triticum aestivum* L.) [16]. The R2R3-MYB gene subfamily is not only numerous in plants, but also diverse in function. R2R3-MYB transcription factors play a more important role in plant growth and development, such as being involved in the regulation of the plant phenylpropanoid metabolic pathway [17], biotic and abiotic stress responses [18], cell morphogenesis and differentiation [19], hormone responses [20], and plant defense [21,22].

Banana (*Musa* spp.) is an important staple food for many people in subtropical and tropical regions. It is rich in protein and carbohydrates as a popular tropical fruit, and banana production significantly contributes to many people’s incomes [23]. The majority of edible cultivated banana originated from intraspecifc or interspecifc hybridization between wild diploid *Musa acuminata* (A genome) and *Musa balbisiana* (B genome) species. The A and B genomes have been assembled with high quality [24,25]. “Cavendish” banana is a triploid (AAA) cultivar banana formed by intraspecific hybridization in A genome, which is the most widely cultivated and commercialized variety due to its resistance to Foc R1 [26,27]. After forming a triploid, asexual propagation is the only way to reach the reproductive mode of the “Cavendish” banana, which leads to low environmental adaptability. Abiotic and biotic stress, such as drought [28], low temperature [29], salt [30], and several devastating diseases [31], affect the yield and quality of the “Cavendish” banana fruit. *Fusarium wilt* of banana (FWB) is prevalent in the main banana-producing areas worldwide and is seriously destructive for the banana industry. It is a typical soil-borne fungus disease caused by *Fusarium oxysporum* f. sp. *cubense* (Foc). Specifically, “Cavendish” bananas are highly susceptible to Foc TR4 [32]. Until now, Giant Cavendish tissue culture variants (GCTCVs) have acquired resistance to TR4 through somaclonal variation [33]. “GCTCV-119” is the best Foc TR4-tolerant alternative cultivar for “Cavendish” [27].

In the *Musa acuminata* genome (DH-Pahang), 499 putative MYB genes, including 362 MYB genes and 137 MYB-related genes, have been predicted [24]; however, only a few R2R3-MYB genes have been functionally characterized [34,35]. Although genome-wide analysis of the MYB family in the *Musa acuminata* genome has been carried out previously [36,37], a systematic analysis of candidate *R2R3-MYBs* involved in banana fruit development, ripening, and responses to stress has not been reported. Because the R2R3-MYB TFs have been confirmed to be associated with abiotic and biotic stress tolerance in many plant species, we systematically analyzed the R2R3-MYB TFs in banana to assess their potential relevance to fruit ripening and stress tolerance.

## 2. Results

### 2.1. Identification and Classification of MaMYBs in Banana

After annotating all protein sequences from the *Musa acuminata* genome (DH-Pahang) on the iTAK website [38], we predicted 499 putative *MaMYBs*, including 362 MYB genes and 137 MYB-related genes, and checked the Plant Transcription Factor Database using *AtMYBs* from Arabidopsis (Appendix A). The remaining sequences were assessed for the existence of complete R2R3-MYB domains using the Conserved Domains Database (CDD). Finally, we obtained 280 R2R3-type *MaMYBs*. All *MaMYBs* were randomly distributed on 11 chromosomes and were named *MaMYB001* to *MaMYB277* based on their chromosomal locations. *Ma00_g01590*, *Ma00_g04340*, and *Ma00_g04960* were not anchored on the chromosome and were named *MaMYB278*, *MaMYB279*, and *MaMYB280*, respectively. The protein length of MaMYBs was changed from 163 (MaMYB244) to 600 (MaMYB154) amino acids. EXPASY analysis revealed that the MaMYB protein sequences had large variations in isoelectric point (*pI*) values (ranging from 4.72 to 10.33) and molecular weight (ranging from 19.29 to 65.65 kDa). The detailed characteristics of MaMYB protein sequences are summarized in Appendix A.

### 2.2. Phylogenetic Analysis of the MaMYB Protein Family

To analyze the evolutionary relationships among the MaMYBs, 280 MaMYB proteins were aligned with 124 R2R3-MYB proteins from Arabidopsis [2] and an unrooted phylogenetic tree was constructed using MEGA6 (Figure 1). Based on the topology and robustness of the resulting maximum likelihood phylogenetic tree, we resolved these R2R3-MYB genes into 36 clades (C1–C36). In our classification of the MYBgene family, we also considered the subgroup categories from Arabidopsis [2,3]; the 280 MaMYBs were grouped into 33 clades. It is noteworthy that there are no R2R3-MYB members from *Musa* acuminata genome distributed in the C3 (S3), C18 (S6), and C32 (S25b) clades. We also found that C4 (MaMYB030, MaMYB087, MaMYB090, MaMYB096, MaMYB149, and MaMYB271), C16 (MaMYB040, MaMYB127, MaMYB170, and MaMYB171), and C23 (MaMYB010, MaMYB058, MaMYB060, MaMYB064, MaMYB068, and MaMYB221) clades only contained MaMYBs from *Musa acuminata* genome but did not contain AtMYB from Arabidopsis. Among the phylogenetic trees, C20 (S20) and C10 (S14b) clades were the largest clades, containing 26 MaMYB members, respectively. C8 (S4) contained 20 MaMYBs. In C1 (S8), C5, C6 (S2), C8 (S4), C9 (S1), C10 (S14b), C11 (S10), C17 (S5), C20 (S20), C25 (S13b), C29 (S17b), C33, C34 (S23), and C36 (S21) clades, the number of R2R3-MYB genes was expanded significantly in the *Musa acuminata* genome compared with the homologous genes from Arabidopsis. In the C12 (S11) and C31 clades, the number of MaMYBs was the same as the number of AtMYBs. In the C35 (S23) and C15 (S12) clades, the number of AtMYBs was greater than the number of MaMYBs (Figure 1).

### 2.3. Conserved Amino Acid Residues in the MaMYBs Domain

Using SMART and Clustal W multiple alignment, 115 highly conserved amino acids were identified. The N terminals of these amino acids formed two special domains, R2- and R3-MYB (Figure 2a,b), which were composed of 59 and 56 amino acid residues, respectively. A conserved map of the N-terminal R2R3 domain of MaMYB proteins was drawn using the weblogo website. The repeat regions of R2 and R3 formed three helixes in the DNA binding domain of MaMYBs, forming an HTH structure that was separated by periodic tryptophan residues (W, Trp), usually as a marker of the MYB domain. Among the three helix structures, the third helix was more conservative than the other two. Compared with R2, the second (29) and third (48) tryptophan residues in the R3 repeat region were also conserved, but the first W residue (6) in the R3 repeat region in most of MaMYB proteins was replaced by F (Phe), I (IIe), and L (Leu) residues (Figure 2a,b). In the R2 domain, W-5\51, E (Glu)-9, D (Asp)-10, L-13, G (Gly)-21, W-26\51, R (Arg)-42\48\50, G (Gly)-44, K (Lys)-45, S (Ser)-46, C (Cys)-47, L-49\55, N (Asn)-53, and P (Pro)-57 were highly conserved (>90%); the E (Glu)-9, G-21,C-47, R (Arg)-50, and W-51 sites were the same in all MYB sequences. In R3 domain, E (Glu)-10, G-22\38, W-29\48, I-32, A (Ala)-33, P (Pro)-37, R (Arg)-39, T (Thr)-40, D (Asp)-41, N (Asn)-42\46, and K (Lys)-45 were highly conserved (>90%); the G-22, W-29, and R-39 sites were the same in all MYB sequences. These differences may be related to the different functions of R2 and R3 in R2R3-type MaMYBs.

### 2.4. Chromosomal Localization and Gene Duplication of MaMYBs

Genome chromosomal location analyses revealed that 277 *MaMYBs* were distributed on 11 chromosomes. As shown in Figure 3, on average, 25 members were distributed on each chromosome. Most members were distributed on chromosomes 4 and 6, which contained 32 members. Chromosomes 5 contained 30 *MaMYBs*. Chromosome 1 had the least, containing only 15 *MaMYBs*.

Segmental duplication, tandem duplication, and retrotransposition are three ways to drive gene family expansion [39,40]. Based on the segmental fragment information in the *Musa acuminata* genome [24], 54 collinears, involving 138 *MaMYBs* and representing approximately 49.29% (138 of 280) of the total *MaMYBs*, were located in the syntenic blocks and had been segmentally duplicated (Figure 3 and Appendix A). All collinear *MaMYBs* within the syntenic regions belonged to the same group in the phylogenetic tree (Figure 1). By locating the genes on the chromosome, only two pairs (*MaMYB017*–*MaMYB018* and *MaMYB170*–*MaMYB171*) were identified as tandemly duplicated (Figure 3 and Appendix A). We did not find any retrotransposons. Based on our analysis, we proposed that the expansion of *MaMYBs* was mainly via segmental duplication during the *Musa acuminata* genome evolutionary process. In the *Musa acuminata* genome, four collinear regions of ancestor block 12 were located on chr05, chr06, chr07, and chr10. Among segmentally duplicated genes, seven *MaMYBs* (*MaMYB120*, *MaMYB127*, *MaMYB130*, *MaMYB166*, *MaMYB169*, *MaMYB170*, and *MaMYB253*) had collinearity and belonged to ancestor block 12 (Figure 3 and Appendix A). However, in the phylogenetic tree, these seven *MaMYBs* fell into four different clades (C9, C13, C16, and C36) (Figure 1). At the protein sequence level, the similarity of amino acid residues among the seven MaMYBs was only 36.06% (data not shown). These results indicate that collinear *MaMYBs* accumulated more variation in the sequence, which may cause gene function differentiation during the *Musa acuminata* genome evolutionary process.

### 2.5. Expression Profile of MaMYBs during Fruit Development and the Postharvest Ripening Stages

To analyze the expression profiles of *MaMYBs*, RNA-Seq data were derived from the fruit development and ripening stages. We annotated the RNA-Seq data using the *Musa acuminata* genome as the reference genome. We deleted 191 *MaMYBs* with reads per kilobase of transcript per million mapped reads (RPKM) values less than 1 (RPKM < 1) at 0 DAF (days after flowering), 20 DAF, 80 DAF_0 DPH (days postharvest), 8 DPH, and 14 DPH (Appendix A). During fruit development and ripening, the RPKM values of 89 *MaMYBs* were more than 5 (RPKM > 5) in any of the five RNA-seq samples (Figure 4 and Appendix A). Many of these genes were highly expressed (RPKM > 10) at 0 DAF and 20 DAF (Figure 4), including 29 *MaMYBs* at 0 DAF and 42 *MaMYBs* at 20 DAF. *MaMYB014, MaMYB018, MaMYB039, MaMYB050, MaMYB120, MaMYB128, MaMYB176, MaMYB206*, and *MaMYB238* were only highly expressed (RPKM > 30) during fruit development, indicating that these genes play important roles in fruit development.

In the postharvest ripening process, 10, 8, and 5 *MaMYBs* were highly expressed (RPKM > 10) at 80 DAF_0 DPH, 8 DPH, and 14 DPH, respectively (Appendix A). Most *MaMYBs* were not involved in the banana fruit postharvest ripening process. Furthermore, the expression levels of *MaMYB001, MaMYB080, MaMYB177*, and *MaMYB278* were highly and differentially expressed (RPKM > 50) at 80 DAF-0 DPH, 8 DPH, and 14 DPH, indicating that these four *MaMYBs* may play important roles in the postharvest ripening process of banana fruit.

### 2.6. The Expression Level of MaMYBs in Banana Seedlings in Response to Osmotic, Salt, Cold, and Foc TR4 Treatments

To analyze the expression profiles of the *MaMYBs*, RNA-Seq data were derived from banana seedlings in response to osmotic, salt, cold, and Foc TR4 treatments. To present the differentially expressed genes visually and exactly, we filtered out the genes with RPKM values less than 5 in both the control (0 days post-inoculation) and treatments. Low-temperature treatment (4 °C) caused the 28 *MaMYBs* to be differentially expressed (log2 Ratio Cold/Control > 1), among which 18 *MaMYBs* were upregulated and 11 were downregulated (Figure 5 and Appendix A). Treatment with drought (200 mmol•L^−1^ mannitol) caused the 28 *MaMYBs* to be differentially expressed (log2 Ratio Osmotic/Control > 1), among which 18 *MaMYBs* were upregulated and 10 were downregulated (Figure 5 and Appendix A). Treatment with salt (300 mmol•L^−1^ NaCl) caused 15 *MaMYBs* to be differentially expressed (log2 Ratio Salt/Control > 1), including 13 upregulated and 2 downregulated *MaMYBs* (Figure 5 and Appendix A). Inoculation with Foc TR4 resulted in the differential expression of 36 *MaMYBs* (log2 Ratio 2 DPI/0 DPI > 1), including 25 upregulated and 9 downregulated *MaMYBs* (Figure 5 and Appendix A).

In the above osmotic, salt, cold, and Foc TR4 infection treatments, we detected 63 differentially expressed *MaMYBs* in banana seedlings, indicating that most of the *MaMYBs* were not involved in the response to various abiotic and biotic stressors. A total of 51 genes were involved in banana’s response to osmotic, salt, and cold stress. *MaMYB031, MaMYB163, MaMYB164,* and *MaMYB191* were differentially expressed in all three stresses, suggesting that these *MaMYBs* may play an important role in banana’s response to abiotic stress. Interestingly, 23 differentially expressed genes (more than half) in the Foc TR4 treatment were also differentially expressed in osmotic, salt, and cold stress (11, 8, and 14 genes, respectively). *MaMYB197* and *MaMYB278* were differentially expressed in banana’s response to osmotic, salt, cold, and Foc TR4 infection processes, indicating that *MaMYB197* and *MaMYB278* may play important roles in banana’s response to multiple stress processes.

### 2.7. Weighted Gene Co-Expression Network of MaMYBs

Weighted co-expression network analysis (WGCNA) is a bioinformatics method that describes the pattern of gene association and can quickly extract gene co-expression modules related to sample characteristics [41]. WGCNA presents the global expression of genes in samples by modular gene classification, and the genes with known functions can be used to predict the functions of genes with unknown functions, providing clues for follow-up biological experiments [42].

To explore the functions, 280 *MaMYBs* were selected as “guide genes” to seek co-expressed genes using an RNA-Seq dataset from 11 different transcriptomes, including those at fruit development and ripening stages, banana seedling responses to osmotic, salt, and cold treatment, and banana roots inoculated with Foc TR4 (Appendix A). A total of 6118 genes as the “target node”, whose expression patterns were closely correlated with 77 *MaMYBs*, were identified with weighted values larger than 0.5 (Appendix A).

Following visualization using Cytoscape [43], the co-expression network of *MaMYBs* was divided into five modules. Modules I–V (blue, red, yellow, green, and purple) contained 69, 2, 4, 1, and 1 *MaMYBs*, respectively (Figure 6). Among these *MaMYBs*, *MaMYB238* had the most association genes, which were associated with 3077 genes (Appendix A). *MaMYB138* and *MaMYB263* had the least associated genes, and only two genes were associated with it (Figure 6). In our results, most of the *MaMYBs* were concentrated in the same module (66 *MaMYBs* distributed in Module I, 89.61% of the total) (Figure 6), indicating that these *MaMYBs* participate in similar biological processes in banana.

Among the *MaMYBs* in the WGCNA network, 20 *MaMYBs* were differentially expressed in banana’s response to Foc TR4, accounting for about 25.97% of the total. We selected seven *MaMYBs* (RPKM > 100 at 0 DPI) and their target node genes for Kyoto Encyclopedia of Genes and Genomes (KEGG) pathway enrichment analysis. Three KEGG pathways were enriched in modules of all seven *MaMYBs*, mainly including metabolic pathways (mus01100), biosynthesis of secondary metabolites (mus00999), and carotenoid biosynthesis (mus00906). Plant hormone signal transduction (mus04075) and alpha-Linolenic acid metabolism (mus00592) pathways were enriched in the *MaMYB059, MaMYB080, MaMYB082, MaMYB097, MaMYB222*, and *MaMYB259* modules, suggesting that *MaMYBs* had functional redundancy in regulating the expression of downstream target genes. The plant–pathogen interaction (mus04626) pathway was enriched in modules of *MaMYB059, MaMYB082, MaMYB097, MaMYB222*, and *MaMYB259* (Appendix A). These *MaMYBs* may be associated with disease-resistance-related genes and participate in banana’s response to Foc TR4 infection.

### 2.8. Expression Patterns of MaMYBs during Banana Seedlings Interacting with Foc TR4

*MaMYB015, MaMYB059, MaMYB080, MaMYB082, MaMYB097, MaMYB197, MaMYB222, MaMYB259*, and *MaMYB278* were highly expressed (RPKM value > 100) at 0 DPI and were selected for quantitative real-time (qRT)-PCR analysis of their expression patterns in the response of the resistant (“GCTCV-119”) and susceptible (“Cavendish”) banana cultivars to Foc TR4 infection (Appendix A). RNA was extracted from the roots of two cultivars at 2, 4, and 6 days post inoculation with Foc TR4 (DPI) and subjected to quantitative RT-PCR (qRT-PCR) analysis. A mock treatment (0 DPI) was carried out using Hoagland’s solution as a control.

In “Cavendish” banana, the expression patterns of nine *MaMYBs* were similar and were downregulated or not differentially expressed at most time points. *MaMYB082* and *MaMYB222* were significantly upregulated at 4 DPI and reached a relative expression level of 1.35-fold at 4 DPI (Figure 7). In “GCTCV-119” banana, *MaMYB015, MaMYB059, MaMYB080, MaMYB082, MaMYB097, MaMYB222, MaMYB259*, and *MaMYB278* were increased at most time points. *MaMYB015*, *MaMYB059*, *MaMYB080*, *MaMYB082*, and *MaMYB259* reached the highest relative expression levels of 3.07-, 7.29-, 5.02-, 5.49-, and 2.49-fold at 4 DPI, respectively (Figure 7). *MaMYB097* reached the highest relative expression level of 2.24-fold at 6 DPI. The expression patterns of *MaMYB222* and *MaMYB278* were relatively consistent and significantly increased at 2 and 6 DPI, and then returned to normal levels within 4 DPI. *MaMYB197* was significantly downregulated at all time points (Figure 7). In summary, these results indicated that *MaMYB015, MaMYB059, MaMYB080, MaMYB082, MaMYB097, MaMYB222, MaMYB259*, and *MaMYB278* were involved in banana’s resistance to Foc TR4 infection in the “GCTCV-119” cultivar.

### 2.9. Verifying the Interaction between MaMYB059 and Promoters of MaRIN4 and MaPR1 in WGCNA Using Yeast One Hybrid

We extracted the co-expression network of *MaMYB059* and found 652 genes contained in the network (Figure 8a). There were 13 genes enriched in the plant–pathogen interaction (mus04626) pathway in the *MaMYB059* module (Appendix A). To further verify the association in the co-expression network, we further identified whether the *MaMYB059* had a relationship with these genes. *MaPR1* (Ma02_t15060.1) and *MaRIN4* (Ma03_t12680.2) were subsequently verified using yeast one hybridization assay and annotated as pathogenesis-related protein 1 C and RPM1-interacting protein 4, respectively, in *Musa acuminata* genome. The −2000 bp upstream sequence of *MaPR1* and *MaRIN4* were obtained from “Cavendish” banana by RT-PCR based on the sequence of the *Musa acuminata* genome. The 1791 and 1471 bp promoter sequences were obtained using transcriptional active site prediction, and the main regulatory elements of *MaRIN4* and *MaPR1* promoters were predicted online by the plantcare. There were three conserved MYB elements (+544, CAACCA, +1029, CAACTG, and −1132 bp, CAACTG) and Myb-binding site (−1154 bp, CAACAG) that bound with MYB TFs in the *MaPR1* and *MaRIN4* promoter sequences. The pGADT7-MaMYB059 vector was used as effectors and the pAbAi vectors carried the *MaRIN4* and *MaPR1* promoters as reporters, respectively. The Y1HGold yeast co-transformed with *MaPR1* promoter-AbAi and pGADT7-MaMYB059 plasmid, *MaRIN4* promoter-AbAi, and pGADT7-MaMYB059 plasmid grew normally on the SD/-Leu medium containing 300 ng/mL AbA. Yeast carrying *MaPR1* and *MaRIN4* promoter-AbAi and pGADT7 did not grow as a control (Figure 8b), demonstrating that *MaMYB059* definitely bound with the sequence of the *MaRIN4* and *MaPR1* promoters, respectively, in yeast. In the dual luciferase assay, the *MaRIN4* and *MaPR1* promoter-driven luciferase reporters and CaMV35S-driven *MaMYB059* effector were separately constructed and used to co-infect tobacco leaf epidermal cells by *Agrobacterium*-mediated transformation (Figure 8c). *MaMYB059* was found to activate the activity of the *MaRIN4* and *MaPR1* promoters, with an LUC/REN ratio of higher than 4.46- and 2.64-fold that of the empty vector (pGreen II 62-SK/pGreen II 0800-LUC), respectively (Figure 8d), indicating that *MaMYB059* is an activation effector of *MaRIN4* and *MaPR1* gene expression.

## 3. Discussion

MYB transcription factors play a central regulatory role in plant physiological processes and play important roles in plants’ responses to biotic and abiotic stress during growth and development [2]. The characteristics of the number of encoding amino acids, isoelectric point, and molecular weight of the MaMYBs were similar to previous results in Arabidopsis and other plants [2]. In plants, R2R3-MYB transcription factors are the most common, and there were 55 transcription factors included in cucumber [13], 126 in Arabidopsis [3], 102 in rice [5,44], 117 in *Vitis vinifera* [6], 157 in *Zea mays* [9], 205 in cotton [10], 244 in soybean [11], 207 in *Populus trichocarpa* [8], 150 in *Medicago truncatula* [15], and 393 in wheat [16]. In our study, 280 R2R3-MYB were identified in the *Musa acuminata* genome. In the *Musa acuminata* genome, the number of our reports was basically the same as that reported by Pucker et al. (285 R2R3-MYBs) [36], and far more than that reported by Tan et al. (222 R2R3-MYBs) [37].

Gene duplication is universal across all organisms and plays a key role in driving the evolution of genomes and genetic systems. It is closely related to translocation, insertion, inversion, deletion, and duplication of small fragments, chromosome rearrangement, and fusion after genome-wide duplication events [14]. The number of R2R3 MYB transcription factor families is closely related to gene duplication in different plant species. The genomes of Arabidopsis, rice, and tomato have a large R2R3 MYB transcription factor family, which may be because they all experienced gene duplication events [14,45]. However, cucumber has not experienced a recent gene duplication event, so it has the fewest number (55) of R2R3 MYB transcription factors [13]. Soybean (*Glycine max*) and Chinese cabbage have a large R2R3 MYB transcription factor family, which may be related to their ancestral tetraploids and triploids, respectively [11,12]. The *Musa acuminata* genome underwent three rounds of WGDs (α/β/γ) [46], resulting in a large number of collinear blocks in the A genome. Fifty-four collinear blocks involving 138 *MaMYBs* were segmentally duplicated genes, and two pairs of *MaMYBs* were tandemly duplicated (Figure 3), suggesting that segmental duplication was the main mode of *MaMYBs* expansion in the *Musa acuminata* genome.

In plants, containing highly conserved R2R3-MYB domains is a typical feature of R2R3-MYB genes [2]. In our results, the R2R3-MYB domains were highly conserved in 280 MaMYB proteins and most possessed characteristic amino acids, especially tryptophan residues, which were similar to cucumber [13], *Populus trichocarpa* [7], and Arabidopsis [3]. Generally, the second and third helices (R2 and R3) form a helix-loop-helix (HLH) structure and are bound to DNA [13,47]. There were more conserved amino acid sites in the R2 domain than in the R3 domain, and the first W (Try) amino acid residue in R3 domain was usually replaced by F (Phe), I (IIe), or L (Leu) residue (Figure 2a,b). These results are consistent with those of tomato [14]. Therefore, these findings indicate that the R2R3-MYB domain in the MaMYB proteins is highly conserved and that all members of the MaMYBs are typical R2R3-MYB genes from the *Musa acuminata* genome.

Fruit development and ripening are important processes for banana commercial value. The process of fruit development is the accumulation of various compounds that form the basis of postharvest fruit quality. MYB transcription factors play important roles in regulating the metabolism of fruit development and quality. In apple, the methylation of the promoter of the *MdMYB10* can affect the expression of *MdMYB10* decrease and anthocyanin accumulation decrease in skin [48]. In *citrus*, the *CrMYB73* can regulate citric acid accumulation in fruit [49]. In grapes, the *VvMYB14* and *VvMYB15* can regulate resveratrol and other stilbenes [50]. In our results, about 10.35% of *MaMYBs* were highly expressed (RPKM > 10) during banana fruit development stages (0 and 20 DAF). The expression profile results indicate that *MaMYBs* are significantly involved in fruit development. We found very few *MaMYBs* that were highly expressed (RPKM > 10) in fruit ripening stages (80 DAF-0 DPH, 8 DPH, and 14 DPH) (Figure 4), indicating that most of the *MaMYBs* may be negative regulators involved in the postharvest ripening of banana fruit. In banana, *MaMYB3* (same with *MaMYB136*) can negatively regulate *MabHLH6* and delay banana fruit ripening [35]. In our study, the expression level of *MaMYB136* was lower than 5 (RPKM < 5) (data not shown). These results suggest that most of *MaMYBs* play negative regulatory roles in the banana fruit ripening stage in the form of low expression. Despite this, *MaMYB001, MaMYB080, MaMYB177*, and *MaMYB278* were highly (RPKM > 50) and differentially expressed at 80 DAF-0 DPH, 8 DPH, and 14 DPH, suggesting that these above *MaMYBs* might play important regulatory roles in banana fruit during the postharvest ripening stages.

Many studies have shown that MYBs are involved in tolerance to abiotic stress [2]. Overexpression of *AtMYB44* enhances stomatal closure to confer drought and salt stress tolerance in transgenic Arabidopsis [51]. *MdoMYB121* overexpression in tomato and apple enhances transgenic plants’ tolerance to abiotic stressors, such as salt, drought, and cold stress [52]. Overexpressing *AtMYB20* enhances salt stress tolerance by suppression of the expression of protein phosphatase 2cs as negative regulators of ABA signaling [53]. In our results, 22.50% of *MaMYBs* were differentially expressed in banana seedlings’ response to osmotic, salt, cold, and Foc TR4 infection (Figure 5), showing that these *MaMYBs* were involved in banana’s response to abiotic and biotic stress.

In plants, MYB transcription factors regulate the defense response by regulating the expression of downstream defense genes to prevent the pathogen invasion [54]. *AtMYB30* is a positive regulator of the hypersensitive cell death program in Arabidopsis and tobacco in response to pathogen attack [55]. *OsMYB30* overexpression enhances resistance to *Magnaporthe oryzae* and induces lignification and thickening of sclerenchyma cells by being directly bound to and activating the promoters of *Os4CL3* and *Os4CL5*, which prevent fungi from infecting rice leaves [56]. In upland cotton, GhMYB108 interacts with GhCML11 to participate in the defense response against *Verticillium dahliae* [5]. *PtoMYB115* overexpression in poplar increases resistance to the fungal pathogen *Dothiorella gregaria* [57]. In our results, nine *MaMYBs* were highly expressed (RPKM > 100) and downregulated significantly (log2 Ratio > 1) in “Cavendish” banana inoculated with Foc TR4 (Figure 5 and Appendix A). WGCNA can identify function-related or similar gene expression modules in high-throughput data and consider gene function and its relationship with biological function as a whole, which can specifically screen the co-expression modules with high biological significance with the target traits [42]. In our results, 27.50% of *MaMYBs* had a co-expression network, suggesting that *MaMYBs* may play important roles in fruit development and ripening and stress responses of banana. “GCTCV-119” is a highly Foc-TR4-resistant mutant in cultivated varieties belonging to the Cavendish group (AAA) and was screened based on tissue culture variation from Giant Cavendish [27,33]. Through qRT-PCR analysis, eight *MaMYBs* were shown to be significantly activated in resistant varieties, except for *MaMYB197*.

In plants, the PR protein shows potential antibacterial activity in vitro, and its accumulation is closely related to resistance during pathogen infection [58]. RPM1-interacting protein 4 is a guard molecule that interacts with the RPM1 protein. AvrRpm1 and AvrB induce RIN4 phosphorylation, leading to a cell environment conducive to pathogen growth. RIN4 is usually a negative regulator of plant defense [59]. Using yeast one hybrid and LUC activity assays, *MaMYB059* bound with the promoter of *MaRIN4* and *MaPR1*, indicating that *MaMYB059* is an activation effector of *MaRIN4* and *MaPR1* expression (Figure 8). These results suggest that *MaMYBs* may be involved in banana response to Foc TR4 infection. Overall, this study proposes new MYB gene candidates for the regulation of diverse biological processes in “Cavendish” banana.

## 4. Materials and Methods

### 4.1. Identification of MaMYB Genes in the Musa Acuminata Genome and Phylogenetic Analyses

To analyze the genome-wide R2R3 MaMYB genes, we first downloaded all gene-coding protein sequences of the *Musa acuminata* genome (A genome) DH Pahang v2 from the Banana Genome Hub (https://banana-genome-hub.southgreen.fr) (accessed on 20 August 2022) [24]. Subsequently, we used the iTAK program16 to identify transcription factors based on the consensus rules, which were mainly summarized within PlnTFDB and PlantTFDB [60,61], and obtained all candidate MaMYB protein sequences. Finally, all candidate MaMYB protein sequences were further examined using BLASTp and CDD (http://www.ncbi.nlm.nih.gov/cdd/) databases in NCBI (accessed on 25 August 2021), and only R2R3 types of MaMYBs were retained. The AtMYB protein sequences from Arabidopsis were acquired from the TAIR (http://www.arabidopsis.org/) database (accessed on 29 August 2021). The full-length MaMYB protein sequences from *Musa acuminata* and Arabidopsis were aligned using ClustalW. Relationships were assessed using maximum likelihood tree with 1000 bootstrap replicates and were created using MEGA 6.0 software [62]. The accession number of the identified MaMYBs is listed in Appendix A. The molecular weight and isoelectric points of the MaMYBs were predicted from the ExPASy database (http://expasy.org/) (accessed on 5 September 2021). The sequence logo for the R2R3MYB domain was created by submitting the multiple alignment sequences to the WebLogo server (http://weblogo.berkeley.edu/logo.cgi) (accessed on 12 September 2021). The KEGG enrichment analysis is carried out in KOBAS (http://kobas.cbi.pku.edu.cn/) (accessed on 27 September 2021).

### 4.2. Chromosome Distribution and Gene Duplications

To determine the physical locations of the *MaMYBs*, the starting and ending positions of 277 *MaMYBs* on each chromosome were obtained from the *Musa acuminata* genome database. Circos (0.63) software (Vancouver, Canada) was used to draw the images of the locations of the *MaMYBs.* Tandem and segmental duplications were also identified according to the plant genome duplication database [63]. Examples of tandem duplication were identified based on physical chromosomal location: homologous *MaMYBs* on a single chromosome, with no other intervening genes, located within 30 kbp of each other, were characterized as tandem duplication [64,65]. Syntenic blocks were detected using MCSCAN (parameters: -a -e 1e-5 -s 5) [25], and all *MaMYBs* located in the syntenic blocks were extracted. Circos (0.63) software was used to draw images of the locations and synteny of the *MaMYBs* (http://circos.ca/) (accessed on 20 October 2021).

### 4.3. Plant Materials and Treatments

“Cavendish” banana (*Musa acuminata* L. AAA group cv. “Cavendish”) is a major trade variety in the world. However, Fusarium wilt seriously affects the production of “Cavendish” banana. Fruit was harvested from the Banana Plantation of the Institute of Tropical Bioscience and Biotechnology (Chengmai, Hainan, 20 N, 110 E). For time point expression analysis, “Cavendish” banana samples at different fruit development stages (0, 20, and 80 DAF) were harvested, which represented fruit developmental stages of budding, cutting flower, and harvest stages, respectively. Fruits stored for 0, 8, and 14 DPH, representing the three progressive ripening stages based on color of the fruit, including green, yellowish green, and yellow, respectively, were selected for postharvest analysis. One-month-old “Cavendish” banana seedlings were grown in Hoagland’s solution under greenhouse conditions [66]. The minimum and maximum temperatures in the greenhouse during the experiment were 25 and 30 °C, respectively, while the relative humidity oscillated between 55 and 80%. For salt and osmotic treatments, “Cavendish” banana seedlings were irrigated with 300 mmol•L^−1^ NaCl and 200 mmol•L^−1^ mannitol for 7 days and the leaves without major veins were harvested for analysis. For cold treatment, “Cavendish” banana plants were maintained at 4 °C for 22 h and the leaves without major veins were harvested for analysis. For the Foc TR4 treatment, we used the susceptible cultivar “Cavendish” banana and resistant cultivar “GCTCV-119” (*Musa acuminata* L. AAA group, cv. “GCTCV-119”) to inoculate Foc TR4. The roots of five-leaf stage Cavendish banana seedlings were dipped in a Foc TR4 spore suspension of 1.5 × 10^6^ conidia/mL. The entire root system was then harvested at 0, 2, 4, and 6 DPI [67]. The above samples were immediately frozen in liquid nitrogen and stored at −80 °C until RNA was extracted for transcriptome analysis and qRT-PCR verification.

### 4.4. Analysis of the Expression Profile of MaMYBs

Plant RNA extraction kit (TIANGEN, Beijing, China) was used to extract total RNAs from samples (Appendix A), and micro spectro photometer and agarose gel electrophoresis were used to detect the quality of total RNAs. RevertAid First-Strand cDNA Synthesis Kit (Fermentas, Beijing, China) was used to convert 3 μg of total RNAs into cDNA. The transcriptome was sequenced on Illumina HiSeq 2000 high-throughput sequencing platform (San Diego, CA, USA). Each sample contained two biological replicates. Gene expression levels were calculated as reads per kilobases per million reads (RPKM) [68]. Differentially expressed genes were identified with a read count of two replicates for each gene (fold change ≥ 2; FDR ≤ 0.001) [69]. A heatmap was constructed with MeV 4.9. All RNA-seq data were uploaded in the CNSA (https://db.cngb.org/cnsa/) of CNGBdb under accession numbers CNP0000292 (CNX0051086, CNX0051087, CNX0051090, CNX0051091, CNX0051097, CNX0051098, CNX0051099, CNX0051103, CNX0051104, CNX0051108, and CNX0051109) and analyzed as described in a previous study [25] (accessed on 20 March 2022).

### 4.5. Weighted Gene Co-Expression Network Analysis

Gene expression patterns for all identified genes were used to construct a co-expression network using WGCNA (version 1.47) [41]. Genes without expression detected in all tissues were removed prior to analyses. Soft thresholds were set based on the scale-free topology criterion employed by Zhang and Horvath [70]. An adjacency matrix was developed using the squared Euclidean distance values, and the topological overlap matrix was calculated for unsigned network detection using the Pearson method. Co-expression coefficients of more than 0.50 between the target genes were then selected. Finally, we extracted the co-expression network of all *MaMYBs* and the network connections were visualized using Cytoscape [71]. We enriched analysis of associated genes using the Kyoto Encyclopedia of Genes and Genomes (KEGG) pathways and chose the sequences of the *Musa acuminata* genome (A genome) DH Pahang v2 as the reference genome.

### 4.6. qRT-PCR Analyses

A total of 0.5 μg total RNAs was extracted from banana roots inoculted with Foc TR4 at 0, 2, 4, and 6 DPI, respectively, and AMV Reverse Transcriptase kit was used to synthesize first-strand cDNA. SYBR^®^ Premix Ex Taq™ (TaKaRa, Dalian, China) and Stratagene Mx3000P (Stratagene, CA, USA) machine were used to analyze MYB gene expression, using cDNA as a template. The thermal cycling conditions were as follows: 94 °C for 3 min, followed by 40 cycles of 94 °C for 15 s, 55 °C for 20 s, and 72 °C for 40 s. The data were analyzed using MxProTM QPCR software (Stratagene, CA, USA). The MaActin transcript (Genebank accession numbers: EF672732) was used as a control. All the primers are listed in Appendix A, and the experiments were carried out with three biological replicates. The differences in C_t_ values between the MaMYBs and MaActin transcripts were expressed as fold changes related to MaActin.

### 4.7. Yeast One-Hybrid Assay

Yeast one-hybrid screening was conducted via the MatchmakerTM Gold Yeast One-Hybrid Library Screening System (Clontech, Dalian, China). The bait fragments (the 1791 and 1471 bp fragments of *MaRIN4* and *MaPR1* promoter, respectively) were cloned into the pAbAi vector. *MaRIN4*-AbAi, *MaPR1*-AbAi, and p53-AbAi were linearized and transformed into Y1HGold to create a bait-reporter strain. Transformants were initially screened on plates containing SD medium without Ura (SD/-Ura) supplemented with 0–1000 ng ml^−1^ aureobasidin A (AbA) for auto-activation analysis.

Full-length coding sequences of *MaMYB059* (Ma03_t29510.1) were cloned into the pGADT7 (AD) prey vector and transferred into the bait-reporter yeast strain. Transformed Y1H Gold was cultured on SD medium with 300 ng ml^−1^ AbA and without leucine (SD-Leu + AbA^300^) at 28 °C for 3 d to test the interaction. pGADT7-Rec (AD-Rec-P53) was co-transformed with the p53-promoter fragment to Y1HGold as a positive control, while AD-empty, MaRIN4-AbAi, and MaPR1-AbAi were used as negative controls. The primers used for the yeast one-hybrid assay are listed in Appendix A.

### 4.8. Dual Luciferase Assay

Full-length coding sequence of *MaMYB059* was inserted into the pGreen II 0029 62-SK vector (SK), and fragments of *MaRIN4* and *MaPR1* promoters were inserted into the pGreen II 0800-LUC vector. All constructs were transformed into *Agrobacterium tumefaciens* GV3101 using the freeze–thaw method [72]. Dual luciferase assays were performed with *Nicotiana benthamiana* grown in a greenhouse for 40 days. *Agrobacterium* cultures were prepared with infiltration buffer (10 mM MgCl_2_, 10 mM MES, and 150 mM acetosyringone) to an OD600 of 0.8. Agrobacterium culture mixtures of TF genes (1 mL) and promoters (100 μL) were infiltrated into the abaxial side of *N. benthamiana* leaves using needleless syringes. Leaves were collected after 2 days of infiltration for luciferase (LUC) and Renilla luciferase (REN) activity analyses using the Dual-Luciferase Reporter Assay System (Promega, Madison, WI) with Modulus Luminometers (Promega). Luciferase activity was analyzed in three independent experiments with six replications for each assay.

### 4.9. Statistical Analysis

Statistical analyses were performed using the Student’s *t*-test. The experimental results were expressed as the mean ± standard deviation (SD). *p* values < 0.05 were considered statistically significant (*), and *p* values < 0.01 were considered highly statistically significant (**).

## 5. Conclusions

We systematically identified 280 *MaMYBs* in the *Musa acuminata* genome and classified them into 33 main groups based on phylogenetic analysis with AtMYBs from Arabidopsis. A total of 277 *MaMYBs* were located on 11 chromosomes. Segmental duplications might have contributed to the expansion of the *MaMYBs*. We analyzed the expression profiles of *MaMYBs* at different stages of fruit development and ripening and found numerous *MaMYBs* that participated in the Cavendish banana response to abiotic and biotic stress. Finally, the co-expression network of *MaMYBs* was constructed using WGCNA to elucidate the *MaMYBs* that might participate in important metabolic biosynthesis pathways in Cavendish banana. The qRT-PCR results suggested that *MaMYB015, MaMYB059, MaMYB080, MaMYB082, MaMYB097, MaMYB222, MaMYB259*, and *MaMYB278* were involved in Cavendish banana resistance to Foc TR4 infection. *MaMYB059* was an activation effector of *MaRIN4* and *MaPR1* gene expression. This comprehensive study improves our understanding of *MaMYBs* associated with fruit development, ripening processes, and stress responses and will establish a foundation for future studies on genetic improvement in Cavendish banana.

## Figures and Tables

**Figure 1 plants-12-00152-f001:**
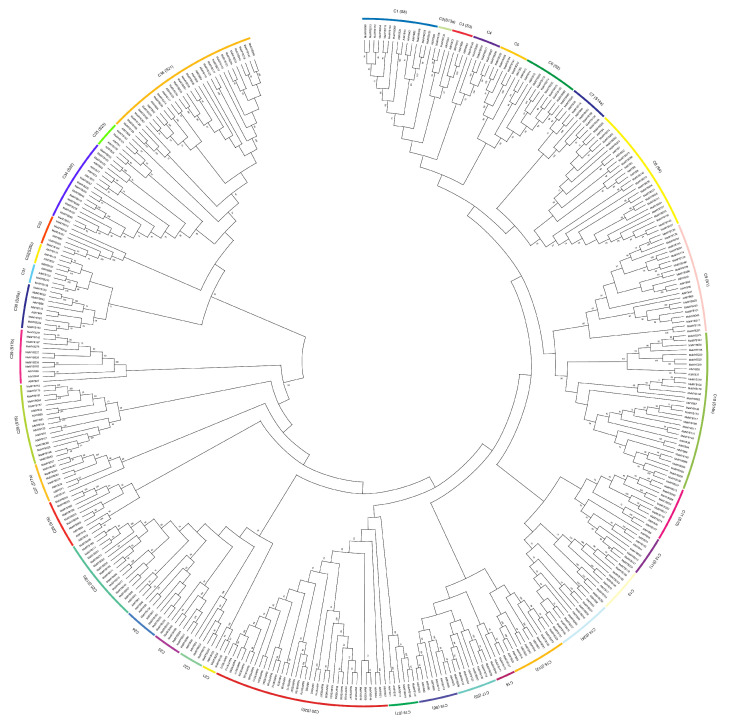
Phylogenetic analysis of R2R3-MYB from Arabidopsis and *Musa acuminata* genomes. The maximum likelihood (ML) tree was drawn using MEGA 6.0 with 1000 bootstrap replicates.

**Figure 2 plants-12-00152-f002:**
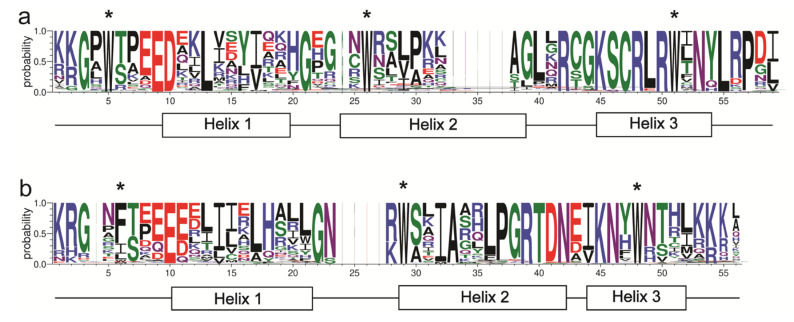
Conservation domain structure analysis of MaMYB proteins. Sequence logos of the R2 (**a**) and R3 (**b**) MYB repeats for the full-length alignment of all R2R3-MYB domains from the *Musa acuminata* genome. The bit score indicates the information content for each position in the sequence. The positions of the three α-helices are marked (Helix 1 to 3). The asterisks (*) indicate the conserved tryptophan residues (W) in the MYB domain.

**Figure 3 plants-12-00152-f003:**
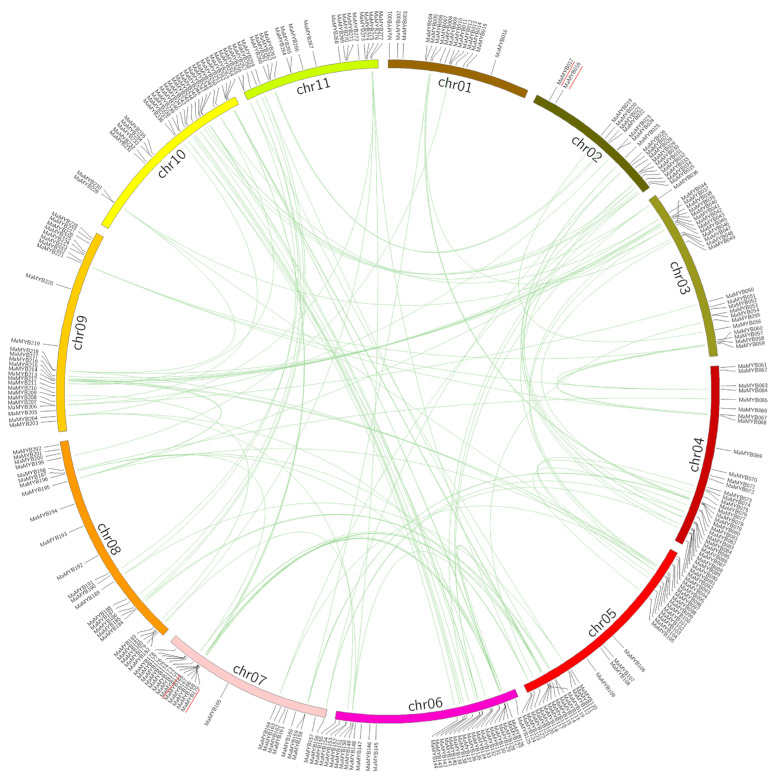
Distribution and synteny analysis of *MaMYBs* on the 11 chromosomes of the *Musa acuminata* genome. The locations of the *MaMYBs* are indicated by vertical black lines. The segmental duplicate *MaMYBs* are connected with green lines, and red lines indicates tandem duplicated *MaMYBs*.

**Figure 4 plants-12-00152-f004:**
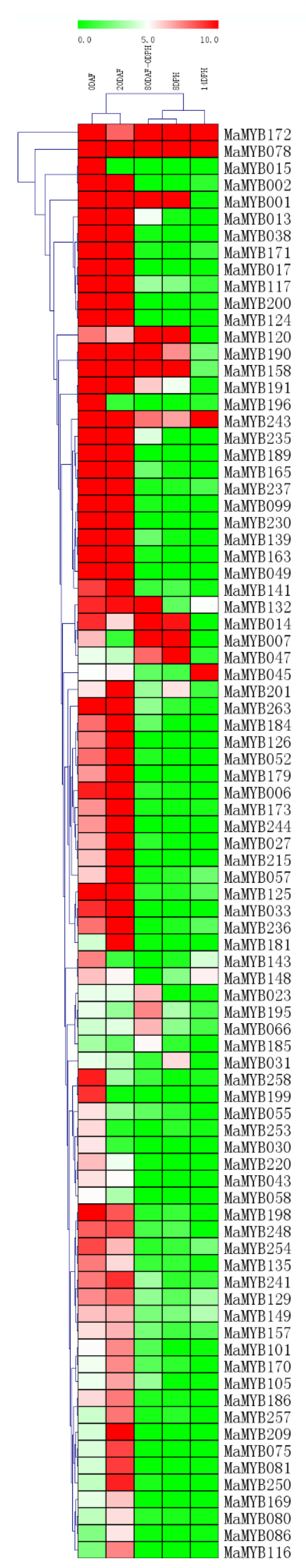
Expression patterns of *MaMYBs* in different stages of fruit development and ripening. The heatmap with dendrogram was created based on the RPKM values of the *MaMYB****s***. Differences in gene expression changes are shown in color as the scale.

**Figure 5 plants-12-00152-f005:**
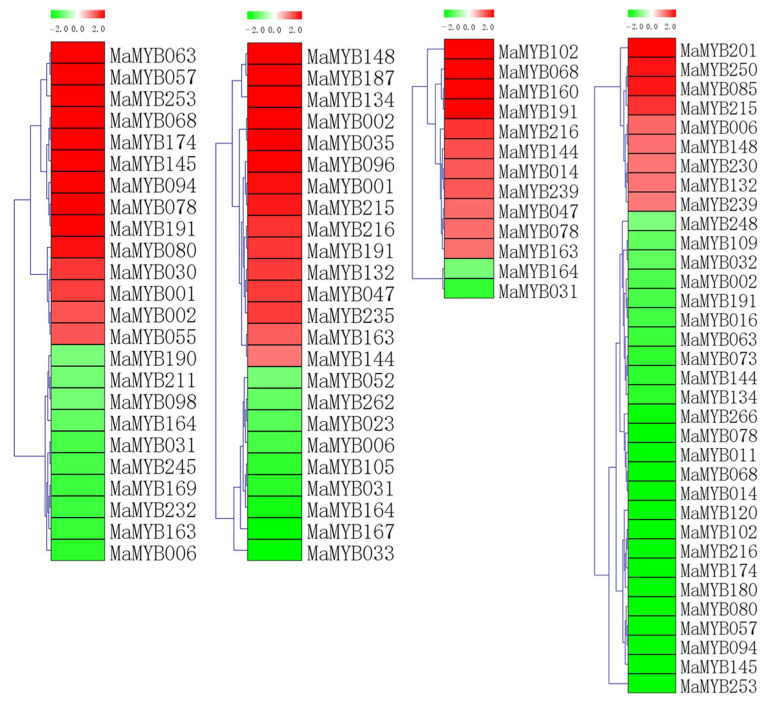
Expression patterns of *MaMYBs* in response to cold (**a**), osmotic (**b**), and salt (**c**) treatments and inoculation with Foc TR4 (**d**) in banana. A Log2-based fold change was used to create the heatmap. Differences in gene expression changes are shown in color as the scale.

**Figure 6 plants-12-00152-f006:**
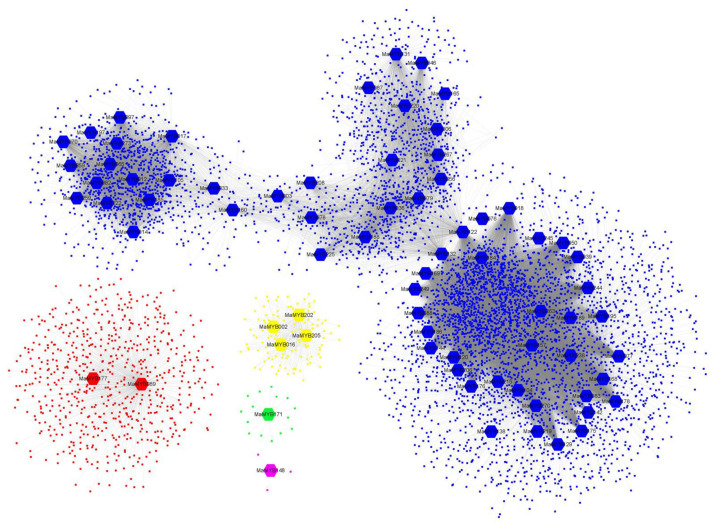
Co-expression network of banana generated using the *MaMYBs* as guides. The network comprises 6118 genes (nodes); the hexagon represents *MaMYB****s***; blue represents module I, red represents module II, yellow represents module III, green represents module VI, and purple represents module V.

**Figure 7 plants-12-00152-f007:**
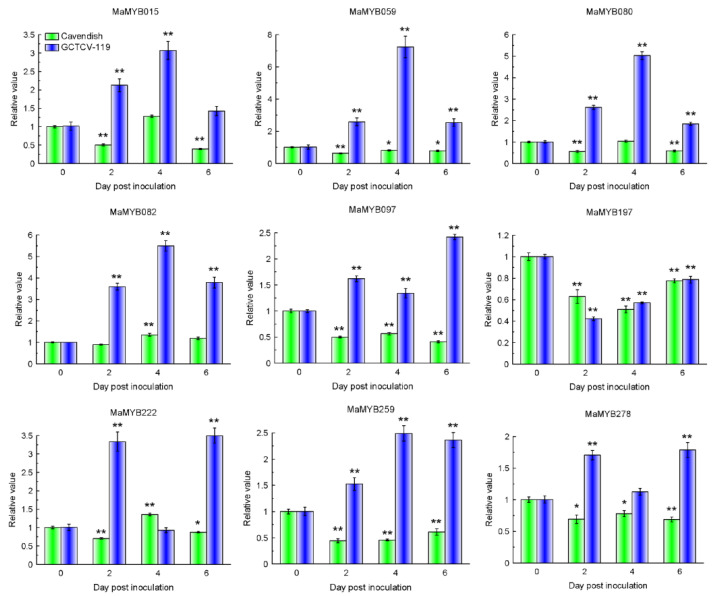
Expression patterns of nine *MaMYBs* in “Cavendish” and “GCTCV-119” banana seedlings inoculated with Foc TR4 identified using qRT-PCR. The results are presented as differential relative transcript abundance. The data represent the mean ± standard deviation (SD) of three replicates. The *y*-axis shows the transcript fold-change relative to that in the control (0 DPI). * and ** indicate significant differences from the control (0 DPI) at *p* < 0.05 and 0.01, respectively.

**Figure 8 plants-12-00152-f008:**
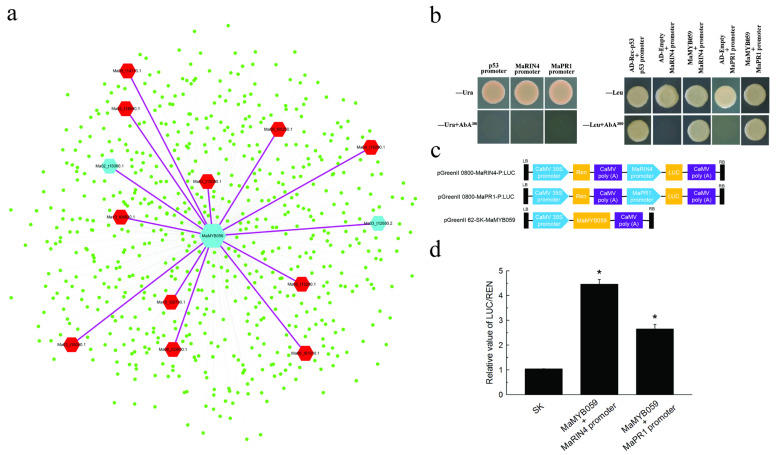
MaMYB059 individually activates the *MaRIN4* and *MaPR1* promoters. (**a**) Co-expression network of *MaMYB059* by WGCNA. (**b**) Yeast one hybrid assay of *MaMYB059* binding with *MaRIN4* and *MaPR1* promoters. (**c**) Schematic diagrams of the effector and reporter constructs used for the dual LUC assay. (**d**) *MaMYB059* activates *MaRIN4* and *MaPR1* promoters in dual-luciferase assays. *p* values < 0.05 were considered statistically significant (*).

## Data Availability

The data presented in this study are available in the article and its Appendix A.

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
