# Peer review of "Genome-Wide Characterization and Analysis of R2R3-MYB Genes Related to Fruit Ripening and Stress Response in Banana (Musa acuminata L. AAA Group, cv. ‘Cavendish’)"

_plants, 2022, doi:10.3390/plants12010152_

Round 1
Reviewer 1 Report
The Ms, Genome-Wide Characterization and Analysis of R2R3-MYB 2 Genes Related to Fruit Ripening and Stress Response in Banana 3 (Musa acuminata L. AAA group, cv. Cavendish), is nicely designed and executed. It can be accepted after suggested revision and inclusion of certain analyses.
1. When MaMYB are written in context to protein, they should not be italic. When
MaMYB are written in context to genes, they should be italicized.
2. Lot of grammatical errors have been done in the manuscript. So, here is a need of
deep revision of every line.
3. Write down the botanical name of plants in the entire manuscript and italicize them.
4. Interaction analysis can be done by using STRING, miRNA interaction can also be included to further strengthen the findings. You may see the reference https://www.mdpi.com/2075-1729/12/7/941
Line no. 19-21 should be rewritten.
5. Line no. 33-36 should be rephrased.
6. Repetitions should be avoided in the whole abstract.
7. Line no. 63- 64 should be rephrased.
8. Line no. 71-73 is not correct scientifically. It should be either removed or recorrected.
9. Line no. 73-74 seems incomplete. Therefore, it should be rewritten.
10. Repetitions are found in line no. 92-93 and 93-94. Therefore, it should be avoided.
11. Clearly mention the number of clades that you have found during phylogenetic analysis.
Variation is found in the number of clades throughout manuscript. It should be corrected.
12. Line no. 171-172, should be rewritten.
13. Figure no. 1 and 3 are not clearly visible.
14. Duplication events and Ka/Ks analysis can also be included. Author may follow suggested Ms https://www.mdpi.com/2223-7747/11/7/911
In the Figure 4, the graph of MaMYB275 is missing.
15. Line no. 348-349 found incomplete. So, it should be rephrased.
16. The manuscript showing the location of 277 MaMYB genes. Mention about the location of
other 3 genes too.
17. Line no. 440 -444 is repeated. It should be either deleted or replaced.
18. Clearly mention about the method used for the construction of phylogenetic tree. Is it
Neighbor-joining or maximum likelihood method?
Author Response
Dear Ms. MilicaČudić,
Many thanks for your valuable suggestions on our manuscript entitled “Genome-Wide Characterization and Analysis of R2R3-MYB Genes Related to Fruit Ripening and Stress Response in Banana (Musa acuminata L. AAA group, cv. ‘Cavendish’)” (plants-2008973). When we submitted our manuscript last time, you informed us that the manuscript was in principle accepted after review. You required us to revise the manuscript to make some editorial changes so that it is as brief as possible and complies with the format instructions. We are very grateful to you for your and reviewers’ valuable suggestions on our manuscript. Based on these comments and suggestions, we have made careful revisions on our manuscript with track changes. There is a point-by-point response to each comments raised, describing exactly what amendments have been made to the manuscript text.
Many thanks and best regards,
WANG Zhuo (E-mail: wangzhuo@itbb.org.cn)
Key Laboratory of Biology and Genetic Resources of Tropical Crops, Institute of Tropical Bioscience and Biotechnology, Chinese Academy of Tropical Agricultural Sciences, Haikou, China
Reviewer 1 #
- When MaMYB are written in context to protein, they should not be italic. WhenMaMYB are written in context to genes, they should be italicized.
Answer: We have checked it.
- Lot of grammatical errors have been done in the manuscript. So, here is a need ofdeep revision of every line.
Answer: We have checked it.
- Write down the botanical name of plants in the entire manuscript and italicize them.
Answer: We have checked it.
- Interaction analysis can be done by using STRING, miRNA interaction can also be included to further strengthen the findings. You may see the reference https://www.mdpi.com/2075-1729/12/7/941
Answer: We have checked it.
Thank you for your suggestion. In this paper, we mainly use the banana transcriptome to construct a co-expression network. We applied current analysis in this paper which also can achieve our purpose.
Line no. 19-21 should be rewritten.
Answer: We have checked it.
- Line no. 33-36 should be rephrased.
Answer: We have checked it.
- Repetitions should be avoided in the whole abstract.
Answer: We have checked it.
- Line no. 63- 64 should be rephrased.
Answer: We have checked it.
- Line no. 71-73 is not correct scientifically. It should be either removed or recorrected.
Answer: We have checked it.
- Line no. 73-74 seems incomplete. Therefore, it should be rewritten.
Answer: We have checked it.
- Repetitions are found in line no. 92-93 and 93-94. Therefore, it should be avoided.
Answer: We have checked it.
- Clearly mention the number of clades that you have found during phylogenetic analysis.
Variation is found in the number of clades throughout manuscript. It should be corrected.
Answer: We have checked it.
- Line no. 171-172, should be rewritten.
Answer: We have checked it.
- Figure no. 1 and 3 are not clearly visible.
Answer: We updated images with higher pixels
- Duplication events and Ka/Ks analysis can also be included. Author may follow suggested Mshttps://www.mdpi.com/2223-7747/11/7/911
Answer: We have checked it.
Thanks for your good suggestion. The focus of this paper is to find functional MaMYB genes. we suppose the current analysis may be enough to explain our results.
In the Figure 4, the graph of MaMYB275 is missing.
Answer: We have checked it.
- Line no. 348-349 found incomplete. So, it should be rephrased.
Answer: We have checked it.
- The manuscript showing the location of 277 MaMYB genes. Mention about the location of
other 3 genes too.
Answer: We have added in the line 107-108 of manuscript.
- Line no. 440 -444 is repeated. It should be either deleted or replaced.
Answer: We have checked it.
- Clearly mention about the method used for the construction of phylogenetic tree. Is it
Neighbor-joining or maximum likelihood method?
Answer: We have checked it.

Reviewer 2 Report
This study identifies and characterizes banana genes encoding R2R3-MYB transcription factors. A search in the banana wild ancestor genome revealed 280 MaMYB genes corresponding to 33 known Arabidopsis MYB clades. The genes were described in terms of their chromosomal localization (in silico) and the pattern of expression during the development and maturation of fruits, as well as in response to various stresses (transcriptome analyses; qRT-PCR). This information helped the authors propose a possible network of MaMYB functioning in fruit development, maturation and post-harvest development, and seedling stress response. In addition, the binding of MaMYB059, whose expression response to Foc infection differentiated between Foc-resistant and Foc-susceptible banana varieties, to target gene promoters, including MaRIN4 and MaPR1, was characterized and confirmed. Given the known regulatory role of many MYB TFs in plant development, defense and adaptation, this study should be useful for both MYB gene research and crop biotechnology.
The manuscript may be accepted for publication in Plants after a minor revision.
Lines 61-63: The species names should be unified. For example, if the authors write “wheat (Triticum aestivum L.)”, then it should be “Arabidopsis thaliana L.”, “Oryza sativa L.”, “cucumber (Cucumis sativus L.)”, “tomato (Solanum lycopersicum L.)”, “Medicago truncatula Gaertn.”, etc.
Lines 66-67: “…biological and abiotic stress…” - “biotic and abiotic stress…” -.
Line 68: “…plant defense mechanisms responses…”
Gene names should be in italics. For example, Lines 101, 102, 105, 118, 119, 411, 416-419, 423, 480, 629, etc.
Line 142: “The N-terminal of these amino acids formed two special domains…”At the N-terminus, these amino acids formed two distinct domains...”
Lines 371-373: “MYB transcription factor played a central regulatory role in plant physiological processes, and played important roles in plants response to biotic and abiotic stresses during growth and development” – “Transcription factors of the MYB family play central regulatory roles in plant physiological processes, as well as important roles in plant response to biotic and abiotic stresses.”
Lines 375-376: “In plants, R2R3-MYB transcription factor was the most common and largest number of transcription factor and included 55 in Cucumber …” – “In plants, the R2R3-MYB transcription factor family is one of the largest and varies in number of members among species, for example, 55 in cucumber …”
Line 391: “…gene duplication events, so it has the smallest number (55) of R2R3 MYB transcription factor family” - “…recent gene duplication events, so it has the fewest members (55) of the R2R3 MYB transcription factor family.”
Lines 392-394: “Soybean (Glycine max) and Chinese cabbage (Brassica rapa ssp. pekinensis) have a large R2R3 MYB transcription factor family, which may be related to their ancient tetraploid and triploid, respectively” – “Tetraploid soybean (Glycine max) and triploid Chinese cabbage (Brassica rapa ssp. pekinensis) have large families of R2R3 MYB transcription factors, which may be associated with the phenomena of both ancient genome-wide polyploidization and recent gene duplication”
Line 407: “L (Tryptophan)” - “W (Tryptophan)”
Line 413: “various materials” - “various compounds” or “various substances”
Line 439: “…through down-regulates the expression…” - “…through suppression of the expression …”
Line 570: “Total RNA was isolated from banana roots before treatment.” However, Figure 7 shows gene expression not only before treatment, but also at other time points (before treatment and 3 times post treatment).
Line 627: “…into 30 main groups…” BUT Line 120: “…the 280 MaMYBs were grouped into 35 clades…” Abstract: "classified into 33 clades". Where is the truth?
Author Response
Dear Ms. MilicaČudić,
Many thanks for your valuable suggestions on our manuscript entitled “Genome-Wide Characterization and Analysis of R2R3-MYB Genes Related to Fruit Ripening and Stress Response in Banana (Musa acuminata L. AAA group, cv. ‘Cavendish’)” (plants-2008973). When we submitted our manuscript last time, you informed us that the manuscript was in principle accepted after review. You required us to revise the manuscript to make some editorial changes so that it is as brief as possible and complies with the format instructions. We are very grateful to you for your and reviewers’ valuable suggestions on our manuscript. Based on these comments and suggestions, we have made careful revisions on our manuscript with track changes. There is a point-by-point response to each comments raised, describing exactly what amendments have been made to the manuscript text.
Many thanks and best regards,
WANG Zhuo (E-mail: wangzhuo@itbb.org.cn)
Key Laboratory of Biology and Genetic Resources of Tropical Crops, Institute of Tropical Bioscience and Biotechnology, Chinese Academy of Tropical Agricultural Sciences, Haikou, China
Reviewer 2#
Lines 61-63: The species names should be unified. For example, if the authors write “wheat (Triticum aestivum L.)”, then it should be “Arabidopsis thaliana L.”, “Oryza sativa L.”, “cucumber (Cucumis sativus L.)”, “tomato (Solanum lycopersicum L.)”, “Medicago truncatula Gaertn.”, etc.
Answer: We have checked it.
Lines 66-67: “…biological and abiotic stress…” - “biotic and abiotic stress…” -.
Answer: We have checked it.
Line 68: “…plant defense mechanisms responses…”
Gene names should be in italics. For example, Lines 101, 102, 105, 118, 119, 411, 416-419, 423, 480, 629, etc.
Answer: We have checked it.
Line 142: “The N-terminal of these amino acids formed two special domains…”At the N-terminus, these amino acids formed two distinct domains...”
Answer: We have checked it.
Lines 371-373: “MYB transcription factor played a central regulatory role in plant physiological processes, and played important roles in plants response to biotic and abiotic stresses during growth and development” – “Transcription factors of the MYB family play central regulatory roles in plant physiological processes, as well as important roles in plant response to biotic and abiotic stresses.”
Answer: We have checked it.
Lines 375-376: “In plants, R2R3-MYB transcription factor was the most common and largest number of transcription factor and included 55 in Cucumber …” – “In plants, the R2R3-MYB transcription factor family is one of the largest and varies in number of members among species, for example, 55 in cucumber …”
Answer: We have checked it.
Line 391: “…gene duplication events, so it has the smallest number (55) of R2R3 MYB transcription factor family” - “…recent gene duplication events, so it has the fewest members (55) of the R2R3 MYB transcription factor family.”
Answer: We have checked it.
Lines 392-394: “Soybean (Glycine max) and Chinese cabbage (Brassica rapa ssp. pekinensis) have a large R2R3 MYB transcription factor family, which may be related to their ancient tetraploid and triploid, respectively” – “Tetraploid soybean (Glycine max) and triploid Chinese cabbage (Brassica rapa ssp. pekinensis) have large families of R2R3 MYB transcription factors, which may be associated with the phenomena of both ancient genome-wide polyploidization and recent gene duplication”
Answer: We have checked it.
Line 407: “L (Tryptophan)” - “W (Tryptophan)”
Answer: We have checked it.
Line 413: “various materials” - “various compounds” or “various substances”
Answer: We have checked it.
Line 439: “…through down-regulates the expression…” - “…through suppression of the expression …”
Answer: We have checked it.
Line 570: “Total RNA was isolated from banana roots before treatment.” However, Figure 7 shows gene expression not only before treatment, but also at other time points (before treatment and 3 times post treatment).
Answer: We have checked it.
Line 627: “…into 30 main groups…” BUT Line 120: “…the 280 MaMYBs were grouped into 35 clades…” Abstract: "classified into 33 clades". Where is the truth?
Answer: We have checked it.
Round 2
Reviewer 1 Report
I could see any suggested work done. The authors have mostly ignored the suggestions. Further, The co-expression network seems to be of no use. If you go for co-expression, there should be a certain cutoff point with a limited interaction appearance so that one can conclude. Here, nothing is visible in the figure. At least important sections should be highlighted. Earlier suggested interactions should also be part of the study as per my opinion..
Author Response
Dear reviewer,
Thank you for your suggestion again.
STRING and miRNA interaction analysis are not included in our experiment plan, so it is impossible to complete the experiment supplement in a short time. In addition, the co expression network constructed by WGCNA method is effective. Based on the data of 11 banana transcriptome, it is an effective method to search for banana Hub gene by using WGCNA method to study the correlation between genes in terms of expression amount. STRING network is mainly used in the study of protein-protein interaction. At the beginning of this study, we compared the advantages and disadvantages of STRING and WGCNA. Cavendish banana is a triploid banana cultivar, It is an important method to identify key genes using transcriptome technology. The purpose of our study is to find active or biologically functional MaMYB genes from transcriptome data. Therefore, we give priority to identifying key MYB genes from banana transcriptome data.
Round 3
Reviewer 1 Report
Ms is good. I leave the decision to the wisdom of the editor.
Author Response
Dear reviewer,
Thanks for your comments.